# Orientation Control for Nickel-Based Single Crystal Superalloys: Grain Selection Method Assisted by Directional Columnar Grains

**DOI:** 10.3390/ma15134463

**Published:** 2022-06-24

**Authors:** Songsong Hu, Yunsong Zhao, Weimin Bai, Yilong Dai, Zhenyu Yang, Fucheng Yin, Xinming Wang

**Affiliations:** 1School of Materials Science and Engineering, Xiangtan University, Xiangtan 411105, China; songhu@xtu.edu.cn (S.H.); baiweimin@xtu.edu.cn (W.B.); fuchengyin@xtu.edu.cn (F.Y.); 2Key Laboratory of Advanced High Temperature Structural Materials, AECC Beijing Institute of Aeronautical Materials, Beijing 100095, China; yunsongzhao@163.com (Y.Z.); xuanshangyiyi@163.com (Z.Y.)

**Keywords:** Ni-based single crystal superalloy, directional solidification, selection grain, columnar grains, crystal orientation

## Abstract

The service performance of single crystal blades depends on the crystal orientation. A grain selection method assisted by directional columnar grains is studied to control the crystal orientation of Ni-based single crystal superalloys. The samples were produced by the Bridgman technique at withdrawal rates of 100 μm/s. During directional solidification, the directional columnar grains are partially melted, and a number of stray grains are formed in the transition zone just above the melt-back interface. The grain selected by this method was one that grew epitaxially along the un-melted directional columnar grains. Finally, the mechanism of selection grain and application prospect of this grain selection method assisted by directional columnar grains is discussed.

## 1. Introduction

Nickel-based single crystal superalloys have excellent high-temperature mechanical properties and are the preferred materials for aero-engine and gas turbine blades [1]. Due to the anisotropy of mechanical properties, the single crystal blade has the best service performance when the <001> direction of the crystal is consistent with the maximum stress direction [2]. Directional solidification investment casting, combined with grain selection or seeding technology, is the main method for preparing single crystal blades [3,4]. The grain selection method is the earliest used to prepare single crystal blades, and it is also the most common method for the preparation of single crystal blades in industrial production. In this method, a spiral selector is usually set at the bottom of the casting. A large number of grains is nucleated firstly at the bottom of the spiral selector, and then the competitive grain growth occurs during the directional solidification process, and finally, only one grain survives at the outlet of the spiral selector [5].

The spiral selector can generally be divided into two parts: the starter block and the spiral part [6,7,8]. The functions of these two parts have received extensive attention and a consensus has been reached [5,7,9]. The main function of the starter block is obtaining well-oriented grains with the <001> texture [10]. Furthermore, the main function of the spiral part is to accelerate the competition grain’s grain and ensure only one grain is close to the inner side of the spiral channel growing into the casting cavity [11]; therefore, the orientation of the final surviving grain is mainly controlled by the competitive grains growth process in the starter block [12]. Only the primary orientation of the surviving grain can be limited within a certain range, which is the degree of <001> direction deviating from the directional solidification, whereas the second orientation is randomly distributed due to the selection mechanism.

In order to improve the control level of the crystal orientation of the surviving grain obtained by the grain selection method, researchers have carried out related research from multiple perspectives, such as the competitive grain growth [13,14], the optimization of the starter block structure and size [10,12], and the effect of process parameters on grain selection [3,15]. Zhang and Zhou et al. found that a high withdrawal rate is beneficial for obtaining a sharper <001> texture and smaller grain size attributing to the competitive growth of a large amount of randomly oriented grains [16]. Wang et al. reported that the misorientation of the final grain can be decreased with the increase in the ratio of length to diameter of the starter block [10]. Rezaei et al. reported that the withdrawal rate, height of the starter block, and temperature gradient during directional solidification were inter-dependent and careful optimization of these process parameters was indeed needed to achieve a minimum misorientation of the final grain [17]. Some studies have also shown that by adjusting the structure and size of the spiral part carefully, the orientation of the final grain can also be optimized within a small scale [5,9,12]. At present, the misorientation of the final grain obtained by the grain selection method can be controlled within 15° for turbine blade applications in industrial production.

With the complexity and thinning tendency of turbine blades, the service performance of the single crystal blade is increasingly sensitive to the crystal orientation [18,19]; therefore, the misorientation of the final grain obtained by the grain selection method is required to be narrower. To address this, an adjusted grain selection method assisted by directional columnar grains for single crystal blades is proposed [20]. A directional columnar grain is set in the starter block during directional solidification, which can significantly reduce the misorientation of the final grain. Using Optical microscope and EBSD technology, the preparation process of the adjusted grain selection method assisted by directional columnar grains for a single crystal sample was investigated. It provides a new insight for optimizing the orientation control level of the grain selection method.

## 2. Experiment

The materials used in this paper were DZ125 and DD6, and their nominal composition are listed in Table 1. The cylinder-shaped directional columnar grains used in this work had a diameter of 10 mm and a length of 40 mm. The spiral part used in this study was a spiral part for the industrial production of single crystal blades. The directional columnar grains were plugged into a mold and the clearance between the columnar grains and the mold was less than 0.05 mm. Then, the mold with the directional columnar grains (as shown in Figure 1) was mounted on a water-cooled copper chill plate in a Bridgman furnace. The furnace chamber was evacuated to a partial pressure above 10^−2^ Pa, and then the mold was heated to 1520 °C for 5 min by a graphite heating element. Melton alloy was poured into the preheated mold cavity at 1500 °C and stabilized for 5 min. Finally, the mold was withdrawn to a cold element at a rate of 100 μm/s. During directional solidification, the main heat flow direction is consistent with the withdrawal direction, whereas it is the opposite for the directional solidification; they are all parallel to the columnar axis.

The microstructure of the directional columnar grains used in this study is shown in Figure 2. From Figure 2a, it can be seen that the typical dendrite morphology at the cross-section of the directional columnar grains and the arrangement of the dendrites were not completely consistent due to the difference in the orientation of these dendrites. According to the statistical results, the size of the grains in the directional columnar grains in the cross-section was between 0.8 mm and 5.1 mm, and the average primary dendrite spacing was about 304 μm. The maximum deviation angle of the <001> direction of the grains in the directional columnar grains from the columnar axis was not more than 8°. Figure 2b showed the typical microstructure in the longitudinal section; it can be seen that the <001> direction of the grain deviated from the columnar axial within a small angle.

Following directional solidification, the casting was cleared and macro-etched by a solution of 50% HCl and 50% H_2_O_2_ for the macroscopically inspecting grains evolution process. Casting was subsequently machined using wire electro-discharge machining (EDM), and then the samples were polished and etched with a mixture of 14% HNO_3_, 28% HF and 58% H_3_O_8_ to display the microstructure. An optical microscope (Leica DM-4000, Wetzlar, Germany) and SEM (ZEIS SUPRA 55, Germany) equipped with EBSD were employed to obtain the microstructures and crystal orientation information of the samples. In order to accurately display the information of crystal orientation during directional solidification, an orthogonal coordinate system was established. The 0Z direction is parallel to the directional solidification direction, and the X0Y plane is perpendicular to the directional solidification direction. In particular, the 0X direction coincides with the projection of the secondary dendrite of the grain selected by this method on the X0Y plane.

## 3. Results

The macrostructure of the adjusted grain selection process assisted by directional columnar grains is shown in Figure 3a. It can be seen that well-oriented columnar grains were formed in the starter block, as the growth direction of the grains was basically parallel to the directional solidification. The columnar grains in the starter block were divided into two parts by the melt-back interface, namely the un-melted zone, which was located in the lower part, and the re-solidified zone in the upper part. Figure 3b shows a partial magnified view near the melt-back interface. A number of stray grains can be found to form exclusively at the surface of the directional columnar grains. The nucleation of stray grains was a transient phenomenon; it occurred only near the melt-back interface. Since the orientation of stray grains was random, which is attributed to nucleation behavior, only a few stray grains with an orientation advantage can grow up in the competitive growth process with the columnar grains. This phenomenon was consistent with the formation and growth of stray grains at the melt-back region of seeding for preparing single crystal casting. Subsequently, the number of grains in the starter block did not change significantly, whereas the number of grains decreased rapidly in the spiral part. A single crystal can be rapidly selected as the solidification front climbing to the spiral part; its cross-sectional and longitudinal microstructures are shown in Figure 3c,d, respectively. The <001> direction of the grain prepared by this method was closed to the direction of solidification, as shown in Figure 3d. It indicated that the grain selection method assisted by directional columnar grains could effectively prepare a single crystal with a small misorientation.

In order to further clarify the orientation control level of the single crystal castings prepared by the grain selection assisted by directional columnar grains, there were 16 single crystal castings fabricated using this method, and their orientation information is displayed in Figure 4. The <001> direction of the single crystal castings deviated from the directional solidification was all less than 8 degrees. Especially, the proportion of the single crystal casting whose <001> direction deviated from the directional solidification within 5 degrees was more than 50%. It showed that the grain selection assisted by directional columnar grains could more effectively control the orientation of the single crystal component than the traditional grain selection method. 

In order to further clarify the mechanism for obtaining a single crystal casting with a small misorientation using this adjusting grain selection method, the microstructure in the starter block was characterized, and the results are shown in Figure 5. From Figure 5a, it could be found that the starter block could be divided into four parts after directional solidification, namely the heat-affected zone, the mushy zone, the transition zone, and the directional growth zone. The heat-affected zone marked as I in Figure 5a was located at the lowest region of the starter block. From Figure 5b, it could be found that there was no obvious re-melting phenomenon in this zone, whereas the (γ + γ’) eutectic was almost completely eliminated, indicating that the microsegregation was significantly reduced. Mushy zone II was immediately adjacent to the heat-affected zone and there was no clear boundary between the two zones. Due to microsegregation, the (γ + γ’) eutectic reformed in the interdendritic region after directional solidification, as shown in Figure 5c. Transition zone III was clearly separated from the mushy zone by the melt-back interface, as shown in Figure 5a. Figure 5d shows atypical dendrites formed and some stray grains nucleated at the initial directional solidification stage. Finally, the directional growth zone IV could be found at the top part of the starter block, and a typical columnar microstructure was reformed, as shown in Figure 5a,e. It should be noted that Figure 5a displayed that the grain with a larger misorientation was overgrown by one with a small misorientation.

The crystal orientations at different regions of the longitudinal section were measured by EBSD—the results are shown in Figure 6. From Figure 6(b1,b2), it can be seen that the formation of stray grains began to occur at the transition zone just above the melt-back interface. At the same time, no stray grain was found below the melt-back interface. A number of equiaxed grains formed in the middle region of the transition zone, as shown in Figure 6(c1,c2); however, the grains aligned with the matrix (located un-melted region) orientation still dominated the transition zone. After the solidification interface was advanced to the directional growth zone, the nucleation of stray grains was not found by EBSD detection. Comparing Figure 6(b1,b2) and (d1,d2), it could be found that the crystal orientation of gains in this zone was basically the same as that of the matrix, which was consistent with the metallographic observation in Fig. 5. It indicated that the grains in the directional growth zone were mainly exitaxially grown along the un-melted columnar grains. This could explain why the maximum deviation angle of the single crystal obtained by the final selection method was consistent with the maximum orientation deviation of the columnar grains.

## 4. Discussion

As mentioned in the introduction, the grain selection and seeding methods are the main technique for preparing single crystal components [4,5]. The grain selection technique mainly relies on the competitive grain growth in the grain selector to obtain single crystal casting. In order to reduce the misorientation range of single crystal casting, a lot of effort is required to determine the grain selection mechanism [3,7,12], optimization of the selector structure [8,10], and directional solidification perimeter [16,17]; however, the random nature of the grain selection process has not changed, leading to the orientation range of the selected grain in industrial production still being quite large for advanced turbine blades. The nature of the seeding technique is epitaxial growth along an un-melted seed; therefore, high single crystal integrity is required for seeding to avoid solidification defects growing up from the un-melted seeding. At the same time, the process parameters, such as the gap between the seed and mold, the inner surface roughness of the mold, etc., should be strictly limited to avoid the formation of stray grains at the initial withdrawal stage during directional solidification [21,22]. Some studies have also shown that the seed oxidation occurred prior to melt pouring, which gives us a chance to form solidification defects at the withdrawal stage [23]; therefore, the seeding method is often combined with a selector partner between the seed and component to improve the success rate of the single crystal preparation [24].

The grain selection method for Ni-based single crystal blades relies on the competitive grain growth to obtain only one grain at the top of the spiral selector. Walton and Chalmers first proposed a competitive growth model based on the relationship between dendrite growth rate and undercooling [25]. After revealing this model with a schematic diagram by Rappaz [13], it is widely accepted and can explain a large number of experimental phenomena. Since then, great effort has been devoted to the optimization of the model to better predict the competitive growth process [14,26,27,28]. The grain selection process has been successfully predicted as the grain competitive growth model is applied to the temperature field evolution obtained by numerical simulation [6,7,12]. It has been found that in order to stably obtain single crystals with a small misorientation by the grain selection method, the length of the starter block should be very large, which is basically beyond the acceptable range of industrial production [10]. This is due to the fact that the overgrowth rate among the grains becomes very slow as the primary orientation between the grains is close [13]. Recently, we have proposed a grain selection method assisted by short seeding to obtain only one grain with the desired orientation through the competitive growth between the grain epitaxially grown along the seed crystals with other grains nucleated in the starter block [29]. The seeding will inevitably undergo a certain degree of oxidation, and the oxide layer is not easily broken by pouring melt during directional solidification, which causes the melt to be unable to epixially grow along the un-melted seeding; therefore, this method runs the risk of crystal orientation control failure.

As the roles of the starter block and the spiral part in the grain selection process have been clarified [5,7,11], we provide a grain selection method assisted by directional columnar grains to control the crystal orientation within a small misorientation. Using this novel grain selection method, the misorientation of single crystal casting is limited to a small angle. Similar to the seeding method, the upper part of the directional columnar grains was re-melted, whereas the lower part remained un-melted due to the temperature field characteristics during the heating and holding stage [30]. During the directional solidification process, the melt epitaxially grows on the un-melted directional columnar grains so that the <001> direction of the grains at the top of the starter block deviates from the directional solidification within a small range. Similar to the process of the seeding method, a certain number of randomly oriented stray grains are formed just above the melt-back interface, as shown in Figure 5 and Figure 6. These stray grains are overgrown by the grains grown epitaxially along the directional columnar grains in the subsequent directional solidification process. The <001> direction of the grain selected by this method deviates from the directional solidification direction within 8 degrees, such as that shown in Figure 4, which is not greater than the maximum <001> direction deviation angle of the directional columnar grains used in this method. It showed that the final grain selected by this novel method was one that grew epitaxially along the un-melted directional columnar grains. 

The grain selection assisted by the directional columnar grains method not only uses the competitive growth characteristics of the grain selection method to obtain single crystal casting but also takes advantage of the epitaxial growth of the seeding method; therefore, this method has unique characteristics compared to the grain selection and seeding methods, which are as follows: (1) Compared to the grain selection method, as shown in Figure 3, this method can control the orientation deviation within a little range of the single crystal casting by selecting an appropriate directional columnar grain used in this method because of the characteristic of the epitaxial growth. (2) Compared to the seeding method, the microstructure of the directional columnar grains used in this method is easier to obtain than that of the seeding. The nucleation and growth of stray grains can be tolerated due to the competitive growth characteristic, as shown in Figure 5 and Figure 6. The grain selection assisted by the directional columnar grains method can be used as the main method in the transition stage of the grain selection method to the seeding method in actual industrial production for advanced single crystal turbine blades, taking into account product requirements, technical difficulty, and cost issues.

## 5. Conclusions

A grain selection assisted by directional columnar grains for single crystal casting was studied using the Bridgman technique at a withdrawal rate of 100 μm/s. The directional columnar grains partially melt-back during directional solidification, and it was divided into four parts after directional solidification, namely the heat-affected zone, the mushy zone, the transition zone, and the directional growth zone. A lot of stray grain appeared in the transition zone just above the melt-back interface, and almost all of them did not grow up during the directional solidification process due to competitive growth with grains growing epitaxially along the un-melted directional columnar grains. The grain selected by this novel method was one that grew epitaxially along the un-melted directional columnar grains; therefore, the <001> direction of the single crystal casting prepared by this method deviated from directional solidification within a small misorientation. This method can be used for advanced single crystal turbine blades in actual industrial production in the transition stage of the grain selection method to the seeding method.

## Figures and Tables

**Figure 1 materials-15-04463-f001:**
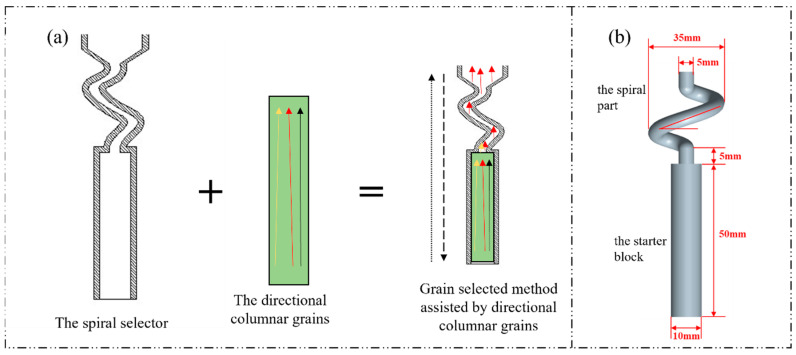
Schematic diagram of the grain selected method assisted by directional columnar grains: (**a**) the novel method; (**b**) the spiral selector used in this paper.

**Figure 2 materials-15-04463-f002:**
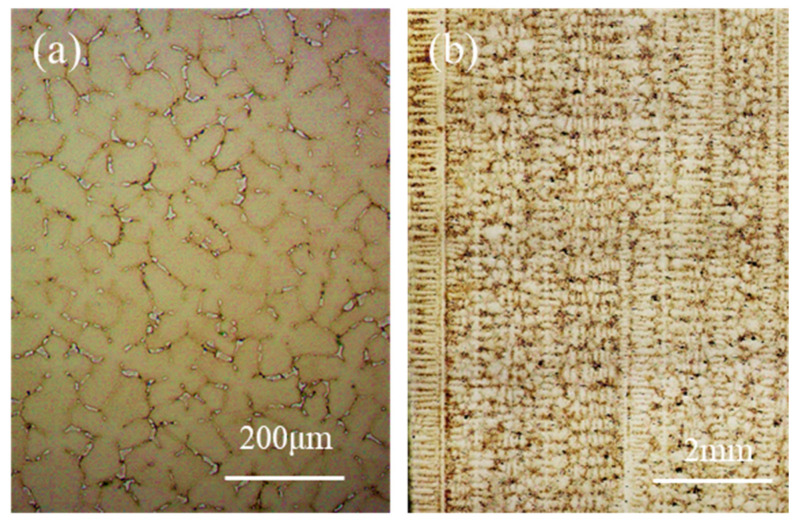
The cross-sectional (**a**) and longitudinal sectional (**b**) microstructure of the directional columnar grains.

**Figure 3 materials-15-04463-f003:**
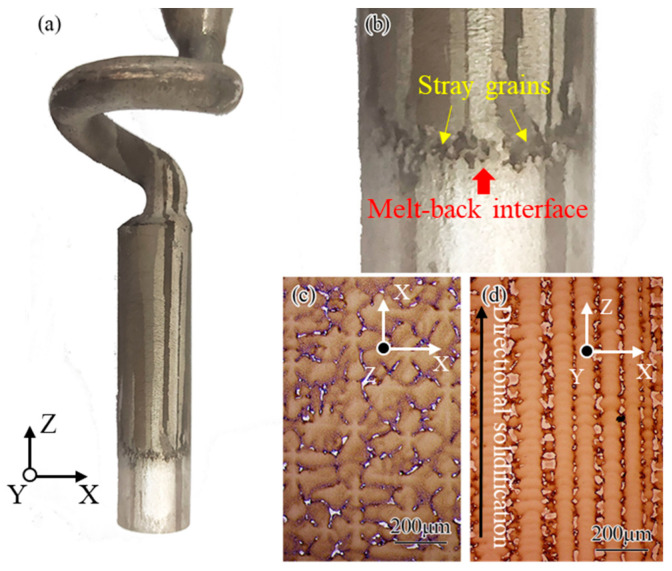
The macrostructure of the adjusted grain selection process assisted by directional columnar grains: (**a**) a magnification view near melt-back region; (**b**) cross-sectional; (**c**) longitudinal (**d**) microstructure of casting.

**Figure 4 materials-15-04463-f004:**
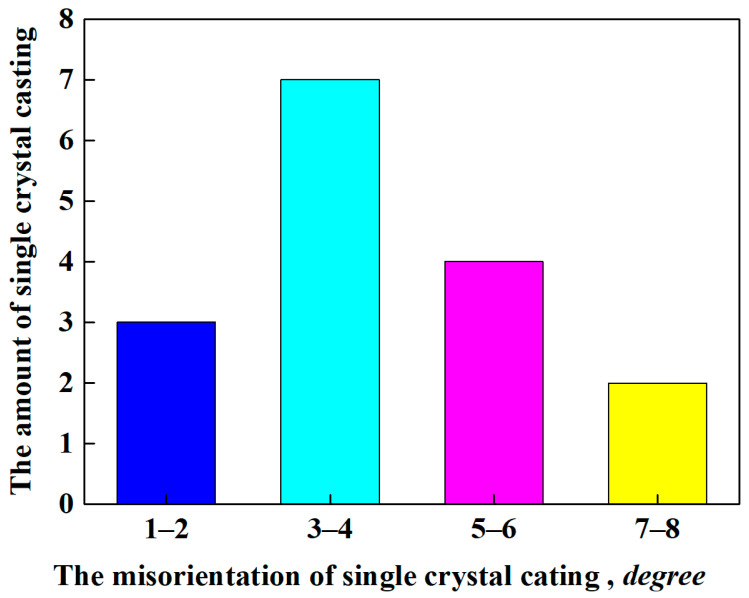
The statistical result of <001> direction of single crystal casting deviated from the directional solidification.

**Figure 5 materials-15-04463-f005:**
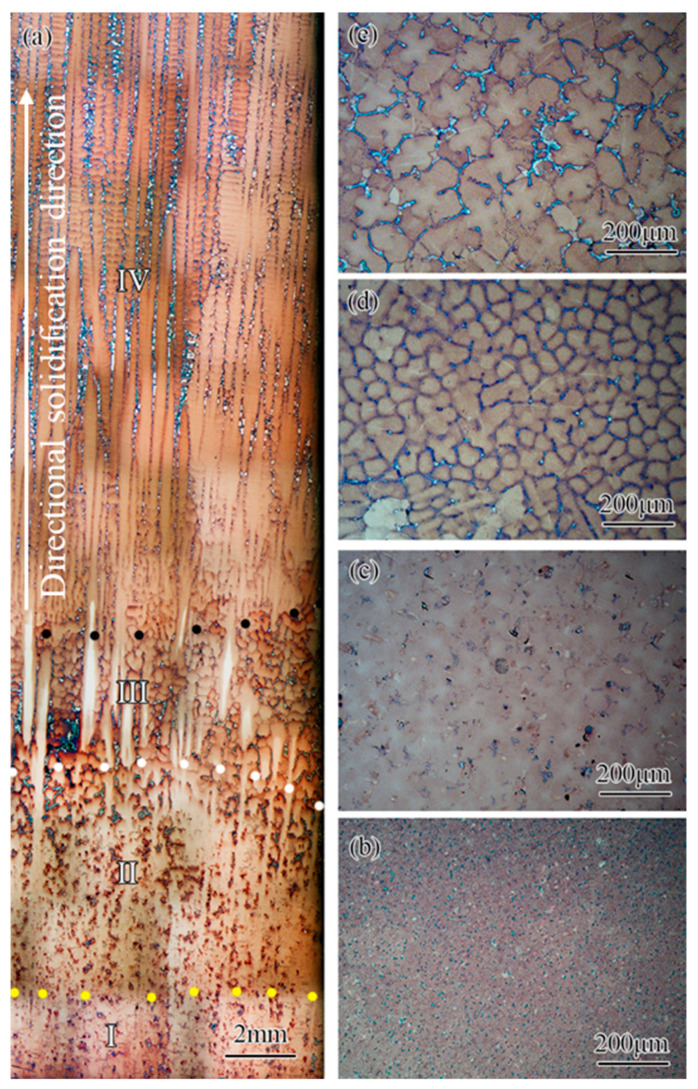
The microstructure of the directional columnar grains after preparing single crystal casting: (**a**) the longitudinal microstructure, and the cross-section microstructures in (**b**) heat-affected zone I; (**c**) mushy zone II; (**d**) transition zone III; (**e**) directional growth zone IV.

**Figure 6 materials-15-04463-f006:**
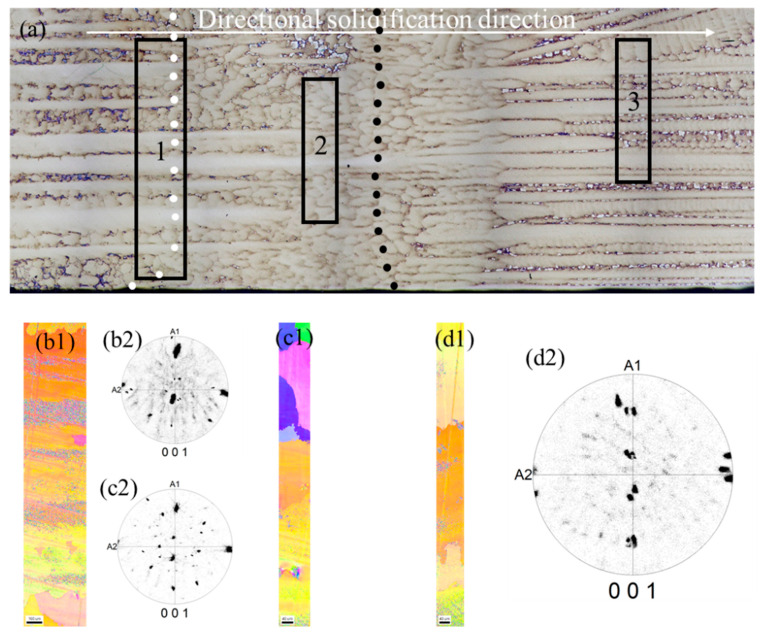
The metallographic microstructure and EBSD results detected by local area: (**a**) the longitudinal microstructure and EBSD maps and corresponding polar diagram tested in (**b1**,**b2**) zone 1, (**c1**,**c2**) zone 2, and (**d1**,**d2**) zone 3 in (**a**).

**Table 1 materials-15-04463-t001:** The nominal compositions of the DZ125 and DD6 (wt.%).

Element	Cr	Co	W	Mo	Re	Al	Ti	Ta	Nb	C	Hf	B	Ni
DZ125	8.9	10.0	7.0	2.0	-	5.2	0.9	3.8	-	0.1	1.5	0.015	Bal.
DD6	4.3	9.0	8.0	2.0	2.0	5.6	-	7.5	0.5	-	0.1	-	Bal.

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
