# Peer review of "Orientation Control for Nickel-Based Single Crystal Superalloys: Grain Selection Method Assisted by Directional Columnar Grains"

_materials, 2022, doi:10.3390/ma15134463_

Round 1

Reviewer 1 Report

I have learned that directional columnar grains could assist grain selection during the processing of single crystal nickel-based superalloys. this is an interesting piece of work and a technically sound manuscript. however, I recommend that the manuscript be subjected to minor editorial corrections before acceptance. This will make the paper reads smoothly.

Some of the edits needed include:

Page 2

Line 12 - benefitted should be changed to beneficial

Line 16 - care should be changed to careful; need to needed.

Line 23 - ......required to be more narrow. Therefore

Line 24-28 should be revised.

EBSD is a metallographic characterization technique. Please revise this statement and be more specific.

Line 31 - table 1 should change to Table 1

Page 3

Explain what a mature spiral part is

Change Form Fig. 2a to From Fig. 2a; change .And to and

Page 4

In the result section, change porcess to process.

The statement - The nucleation of stray grains.... should be revised.

Page 8

In the discussion section, after reference [23], change therefore to Therefore

change angel to angle

Revise the statement... It is indicated that the grain selected.......

Reviewer 2 Report

Authors have studied the orientation control during solidification process of Ni based super alloys to realize preferred crystallographic orientation. For a general reader to follow the purpose of paper and also to interpret the results, authors must provide a detailed schematic illustration of geometry of casting process together with the indication of heat-flow direction and the preferred growth direction and the actual attained direction. With such a schematic authors can clearly define the existing literature and the novel step they introduced. While presenting results too for every presented microstructure, one should give geometrical orientation from which the cross-section is prepared. Without such detail of heatflow direction and the actual cross-section plane it becomes very difficult to follow the presented results. Authors report the remelting of certain orientations due to growth of other specifically orientated grains. Authors may provide scientific reasoning for remelting based on any compositional heterogenity and/or solid/liquid interfacial energy and/or latent heat of fusion overshooting local temperature beyond melting point.

Reviewer 3 Report

I am happy to write to you. In connection with a manuscript entitled; Orientation control for nickel-based single crystal superalloys:grain selection method assisted by directional columnar grains”. This it can be accepted in your valuable journal after doing some major revisions which manuscript can be summarized as the followings:

1.     Abstract must consist of the purpose and methodology along with significant findings, which is lacking in the current article. need to be changed

2.     The authors did not mention the novelty of their work. Thus, the novelty of this work should be clearly discussed.

3.     There are errors in the better language, the language must be correctedز

4.     The conclusion is written very briefly, it must be written extensively

Kind regards    

Reviewer 4 Report

I think that paper entitled “Orientation Control For Nickel-Based Single Crystal Superalloys: Grain Selection Method Assisted by Directional Columnar Grains” is good scientific quality, well organized, achieved results are clear and understandable. I recommended to accept and publish the paper after small revision.

I think that it will be good that authors carefully read the article to avoid some formal errors.

Some of them are listed bellow:

Page 2, paragraph 3, Line 3: written is ”therefore”, correct is “Therefore”

Page 3, paragraph 2, Line 6: Written is: “The maximum deviation angel”,  correct is “The maximum deviation angle”.

Page 5: Especially, the proportion of the single crystal casting whose <001>direction deviated from the directional solidification within 5 degree was more than 50%

Dot in the end of sentence is missing.

Page 6, Fig. 4: caption of axis x. Written is: “crsytal cating”, correct is “crystal casting”.

I think it will be nice to enlarge Fig. 6. There is space available.

Written is: “direction deviation angel”, correct is “direction deviation angle”.

Round 2

Reviewer 2 Report

Authors have addressed my comments and thus i recommend to accept this for publication 

Reviewer 3 Report

Greetings

The authors carefully made all necessary adjustments. Therefore, I recommend that you accept the manuscript.